# Association between triglyceride and depression: A systematic review and meta-analysis

**Di-Ru Xu[1], Xi Gao[1], Li-Bo Zhao[2,3], Shu-Dong Liu[2,3], Ge Tang[2,3], Chan-Juan Zhou[3], Yu Chen[3,4] ***

**1** Department of Dermatology, University-Town Hospital of Chongqing Medical University, Chongqing, China, **2** Department of Neurology, Yongchuan Hospital of Chongqing Medical University, Chongqing, China, **3** Chongqing Key Laboratory of Cerebrovascular Disease Research, Chongqing, China, **4** Department of Neurology, Bishan Hospital of Chongqing Medical University, Chongqing, China

* cheny@hospital.cqmu.edu.cn

**Data Availability Statement:** All relevant data are within the manuscript and its Supporting Information files (S1 Dataset).

**Funding:** National Natural Science Foundation of China (Grant No. 81601207)/. The funder of

## Abstract

Depression is accompanied by dyslipidemia, which may increase the risk of stroke and coronary heart disease. This study sought to quantitatively summarize the clinical data comparing peripheral blood triglyceride (TG) concentrations between patients with major depressive disorder (MDD) and healthy controls (HCs). Studies were searched in PubMed, EMBASE, PsycINFO, and Cochrane Databases up to March 2023. We also reviewed the reference lists of obtained articles. Mean (±SD) for TG concentrations were extracted, combined quantitatively using random-effects meta-analysis, and summarized as a standardized mean difference (SMD). Subgroup analysis and meta-regression was performed to explore the resource of heterogeneity. Thirty-eight studies measuring the concentrations of peripheral blood TG in 2604 patients with MDD and 3272 HCs were included. Meta-analysis results indicated that TG levels were significant higher in patients with MDD than in HCs (SMD = 0.31, 95% confidence interval [CI]: 0.16 to 0.46, $Z_{46}$ = 4.05, $p < 0.01$). Heterogeneity was detected ($\chi^2$ = 269.97, $p < 0.01$, $I^2$ = 85%). Subgroup analysis demonstrated significant differences in TG levels between patients with MDD and HCs depended on age, body mass index and drug use ($p < 0.05$), but no differences between groups. Meta-regression also found no significant variables. TG level was significantly elevated in depression, which may explain the increased risk of cardiovascular and cerebrovascular events in depression.

## 1. Introduction

MDD is a common disease with a 12-month prevalence of 6.6% and a lifetime prevalence of 16.2%. MDD is a main contributor to years of life lived with disability, characterized by mood disturbances, loss of interest in activities, and deficits in cognitive functions [1, 2]. Accumulating evidence indicates that depression can increase the risk of coronary heart disease and stroke [3–5]; however, the underlying mechanism by which depression elevates the risk of cardiovascular and cerebrovascular diseases is unclear. Although treatment of hyperlipidemia

Professor Li Bo Zhao had proofread the manuscript.

**Competing interests:** The authors have declared that no competing interests exist.

mainly focuses on lowering total plasma cholesterol and low-density lipoprotein cholesterol levels, several recent studies have reported the importance of TG levels in atherosclerotic cardiovascular and cerebrovascular diseases [6, 7].

TG is the most abundant lipid in the peripheral circulation. It is a major source of energy and a critical component of the lipoproteins. However, hypertriglyceridemia has been considered as an independent risk factor for atherosclerosis, which leads to myocardial infarction and ischemic stroke in the long term [4, 8]. Many factors, such as BMI, sex, and age, may increase TG concentration. Our previous studies indicated that certain peripheral metabolites [9, 10], including lipid metabolites [11], can differentiate between patients with MDD and healthy subjects with high sensitivity and specificity. Based on these findings, we suppose there may be a significant alteration in TG levels in depressed patients compared with healthy people. Further investigations are needed to identify potential pathophysiological factors and find alternative strategies for treatment.

TG status is most frequently assessed by measuring serum or plasma levels of TG. Peripheral blood TG concentrations have been measured in numerous studies over the past decades. Most of the previous studies [12–15] suggest that depression might be associated with higher concentrations of TG, but a few studies [16, 17] do not support these results, showing any difference between the two groups [18, 19]. The present meta-analysis was performed to compare TG concentrations of depressed patients with that of healthy subjects and explore the modulatory effect of different factors, such as age, sex, body mass index, and antidepressant drugs, in this relationship.

## 2. Methods and materials

### 2.1 Search strategy

The protocol of this study was consistent with the Preferred Reporting Items for Systematic Reviews and Meta-Analyses (PRISMA) guidelines. We searched English language studies using MEDLINE, EMBASE, PsycINFO, and Cochrane databases up to March 2023. The search strategy was as follows: (triglyceride) AND ((((((("depressive symptom") OR "depressed mood") OR "dysthymia") OR "melancholia") OR "major depressive disorder") OR depression). The reference lists of all relevant studies were also searched for any additional trials.

### 2.2 Study selection

Inclusion criteria were: 1) measuring overnight fasting serum or plasma TG concentrations; 2) being a case-control or cross-sectional study 3) Studies that patients only diagnosed MDD on any edition of the Diagnostic and Statistical Manual of Mental Disorders (DSM), Chinese Classification and Diagnostic Disorder (CCMD), or International Classification of Disease (ICD); and 4) inclusion of healthy controls defined as those who are not diagnosed with any disease.

Studies were excluded if they reported depressive symptoms in the context of 1) other neuropsychiatric disorders (e.g., schizophrenia, bipolar disorder, autism etc.), 2) medical illnesses before onset of MDD (e.g., hyperlipemia, dyslipidemia, diabetes mellitus, coronary artery disease, cancer, infection, hepatic disease, etc.), 3) Special physiological conditions (e.g., pregnancy, postpartum or menstrual period, trauma, etc.) or 4) accepting medication that may affect TG levels (e.g., statins, niacin et.).

### 2.3 Data extraction

Each article was separately examined by two independent researchers (Xu and Gao), and any disagreements regarding inclusion were resolved by consensus with a third researcher (Liu).

Serum or plasma TG concentrations (mean ± SD) were converted to mg/dL and extracted for depressed and control subjects. Missing data were requested from the corresponding author. Demographic and clinical characteristics (mean age, body mass index, female percentage, and antidepressant use) and study variables (inclusion criteria, publication date, country and diagnosis method) were extracted.

## 2.4 Quality assessment

The Newcastle-Ottawa Scale (NOS) was used to assess the quality of all studies in this systematic review. Two reviewers (Tang and Zhou) independently assessed the qualities of each study, and the results were compared afterward. Publication bias was assessed using funnel plots and quantitatively measured by Egger's test.

## 2.5 Statistical analysis

Following the Cochrane Handbook for systematic reviews, this meta-analysis was carried out using Review Manager software version 5.3 (http://www.cochrane.org) and Stata software version 15.1 (Stata-Corp, College Station, Texas). Standardized mean differences (SMDs) with 95% confidence intervals (CIs) were calculated for TG levels between MDD and HC. Considering the fact that the TG levels may have been affected by confounding factors, a random effects model was used in statistical analysis [20]. The $I^2$ and Q test statistics were calculated to measure within-study heterogeneity, which was considered significant for an $I^2 > 50\%$ or a P-value $< 0.10$ in the Q test [21]. Subgroup analysis was performed based on medical condition, age and BMI, to identify the source of heterogeneity. Age was stratified by 45 years, because young patient ($< 45$ years) with ST-elevation myocardial infarction (STEMI) has significant higher triglyceride level compared to middle-old age patients ($> 45$) as previous reported [22]. For BMI, stratificational criteria was applied by normal-weight ($> 25$) and overweight ($< 25$). Meta-regression analysis was performed based on demographic characteristics (age, sex, and BMI), diagnosis method (DSM vs. ICD-10 vs. CCMD), publication date, country and drug use to detect the sources of heterogeneity.

## 3. Results

### 3.1 Characteristics of included studies

We identified 3464 studies (2397 records from MEDLINE, 743 records from EMBASE, 118 records from PsycINFO, and 206 records from Cochrane database). After excluding duplicates, we obtained 2618 studies. Then, title and abstract screening resulted in 146 studies whose full texts were checked. Among these studies, 108 studies were excluded because of data unavailability in 33 studies, presence of other diseases in 15 studies, absence of a control group in 23 studies, inconsistency with the inclusion criteria in 26 studies, and involvement of other research in 11 studies (S1 File). Thirty-eight studies were ultimately included and data was summarized in S1 Dataset [12–19, 23–52]. Fig 1 shows the workflow of the searching and selection processes. Notably, 4 studies separately reported TG levels in males and females. Thus, all 42 items were pooled in the meta-analysis. The characteristics of the included studies are summarized in Table 1. An average 7 scores of The Newcastle-Ottawa Scale (NOS) were obtained for included studies after quality assessment. Of those studies, 6 studies had missing BMI data, and only 6 studies explicitly reported TG values in men and women. The included studies contained a total of 2604 patients with MDD and 3272 HCs.

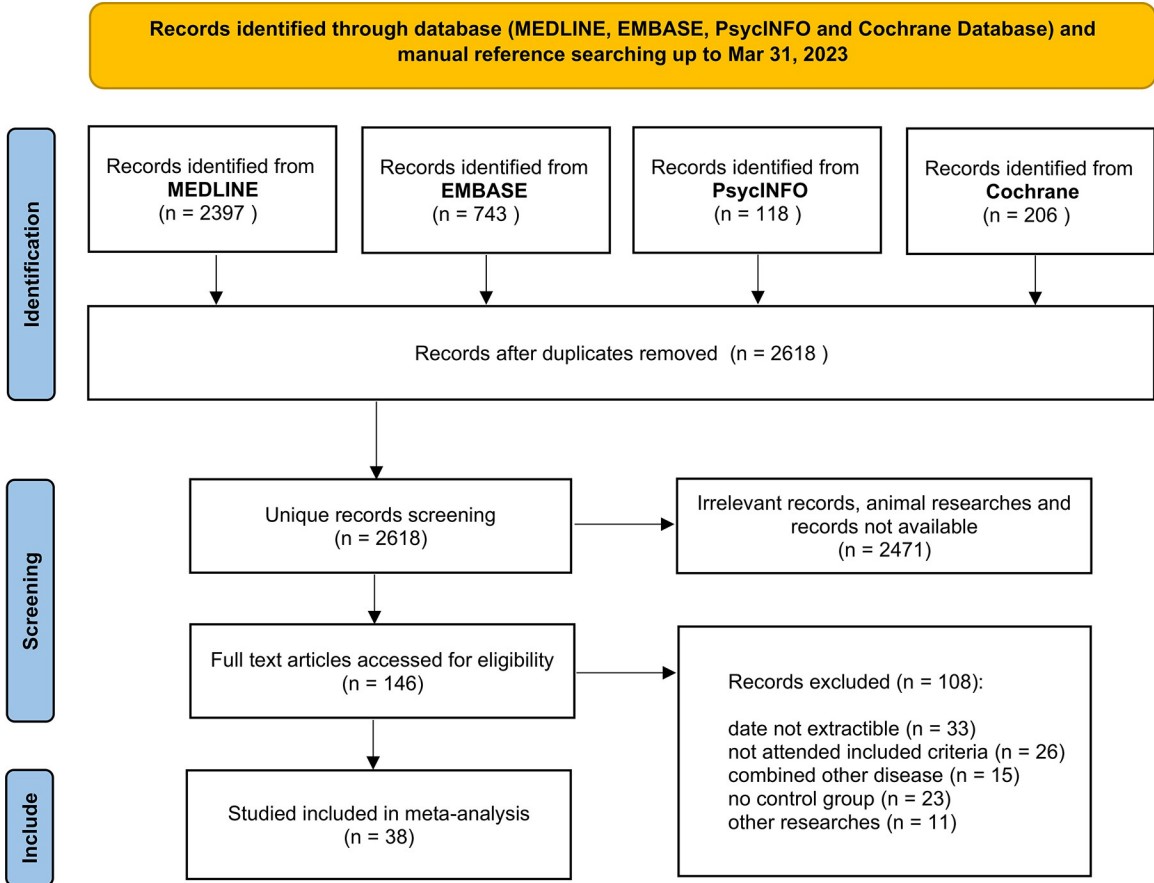

**Fig 1. Workflow of searching and selection.** Searching strategy was manufactured to obtain target studies as many as possible through four main medical databases (MEDLINE, EMBASE, PsycINFO, Cochrane Database).

### 3.2 Triglyceride levels in depressed and healthy subjects

Mean peripheral blood TG concentration was significant higher in patients with MDD compared with HCs (SMD = 0.31, 95% CI: 0.16 to 0.46, $Z_{46}$ = 4.05, $p < 0.01$) (Fig 2). Heterogeneity was detected in this comparison ($\chi^2$ = 269.97, $tau^2$ = 0.2, $p < 0.01$, $I^2$ = 85%); therefore, meta-regression and subgroup analysis were used to identify the potential source of heterogeneity.

### 3.3 Assessment of publication bias

A symmetrical funnel plot was obtained, indicating that there was no publication bias (Fig 3). Similarly, quantitative Egger's test did not show publication bias for triglyceride ($t_{42}$ = 0.39, 95% CI: -2.05 to 3.03, $p$ = 0.699). These results revealed that there was no risk of publication bias.

### 3.4 Subgroup analysis and meta-regression

To better understand the results of the meta-analysis, relevant a priori subgroup analyses were conducted. TG levels were significantly higher in patients with MDD compared to HCs in subgroups of age < 45-year (SMD = 0.28, 95% CI: 0.10 to 0.47, $I^2$ = 87%, $p < 0.01$), age ≥ 45-year (SMD = 0.39, 95% CI: 0.16 to 0.63, $I^2$ = 78%, $p < 0.01$), BMI < 25 (SMD = 0.27, 95% CI: 0.06 to 0.49, $p < 0.01$), BMI ≥ 25 (SMD = 0.43, 95% CI: 0.19 to 0.67, $p < 0.01$), drug-use

**Table 1. Characteristics of included studies and subjects demography.**

| Study(year) | Number (D/C) | Age[a] (D/C) | BMI | Male% | Drug use (%) | Country | Diagnoses | NOS[b] |
|---|---|---|---|---|---|---|---|---|
| Olusi(1996) | 100/100 | 39.58±10.2/39.96±9.8 | 28.1 | 64.0 | 0 | kuwait | ICD-10 | 7 |
| Maes(1997) | 36/28 | 51.1±13.7/47.7±14.2 | n/a | 13.9 | drug-free 10 days | belgium | DSM-III | 8 |
| Khalid(1998) | 28/28 | 38.6±10.7/39.2±10.9 | n/a | 46.4 | 0 | India | DSM-III | 7 |
| Bilici(2001) | 30/32 | 42.2±9.7/42.1±7.4 | n/a | 72.2 | 0 | Turkey | DSM-IV | 7 |
| Sevincok(2001) | 27/24 | 33.29±6.12/33.2±6.78 | 27.4 | 25.9 | drug-free 1 month | Turkey | DSM-III | 8 |
| Huang(2003) | 68/39 | 43.5±14/50.2±11.2 | 23.4 | 45.5 | n/a | China | DSM-IV | 7 |
| Wang(2003) | 50/30 | 30±6.07/28±5.12 | n/a | n/a | 0 | China | CCMD-III | 7 |
| Huang TL(2004) | 68/39 | 43.5±14/50.2±11.2 | 23.2 | 45.5 | n/a | China | DSM-IV | 7 |
| Huang TL(2005) | 109/59 | 31.4±8.5/29.5±4.4 | 21.8 | 29.3 | n/a | China | DSM-IV | 7 |
| Sarandol(2006) | 86/36 | 40.5±10.5/37.2±7.3 | 25.7 | 27.9 | drug-free 3 weeks | Turkey | DSM-IV | 8 |
| Politi(2008) | 25/25 | 53.7±8.4/53.9±9.1 | 25.3 | 48.0 | n/a | Italy | DSM-IV | 8 |
| Cizza(F)(2009) | 77/41 | 35.5±7/35.2±7 | n/a | 0 | 84 | USA | DSM-IV | 8 |
| C. Muhtz (M) (2009) | 8/99 | 47.5±7.3/49..7±11.5 | 26.6 | 100 | 0 | Germany | DSM-IV | 8 |
| C. Muhtz (F) (2009) | 77/41 | 47.7±10.3/49.5±10.6 | 24.5 | 0 | 0 | Germany | DSM-IV | 8 |
| Sagud (2009) | 34/50 | 50.1±6.6/ 44.7±12.8 | 24.2 | n/a | 0 | Kroatia | DSM-IV | 7 |
| PRATIM DAS(2010) | 30/30 | 41.1±10.6/42.0±6.8 | 21.1 | 40.0 | 97 | India | DSM-IV | 8 |
| Lehto(2010) | 88/88 | 49.8±5.7/49.8±8.0 | n/a | 44.3 | 95.3 | Finland | DSM-IV | 7 |
| Aliyazicioglu(2011) | 78/64 | 38±11/28±9 | 23.9 | n/a | 0 | Turkey | DSM-IV | 7 |
| Baghai(2011) | 86/80 | 49.9±13.1/50±13.9 | 24.8 | 61.6 | 43 | Germany | DSM-IV | 6 |
| Hummel(2011) | 65/33 | 50.1±1.7/49.8±14.8 | 25.2 | 35.4 | 0 | Germany | DSM-IV | 7 |
| Ljubicic(2013) | 44/242 | 50.7±12.7/50±12.5 | 27.5 | 63.6 | 0 | Kroatia | DSM-IV | 7 |
| Lamers (2013) | 111/534 | 40.2±12.1/41.3±14.6 | 26.9 | 35.2 | 39.5 | The Netherlands | ICD-10 | 7 |
| Mashele(M)(2013) | 35/52 | 41.9±8.2/43.8±8.2 | 27.6 | 100.0 | 0 | UK | DSM-IV | 8 |
| Mashele(F)(2013) | 46/46 | 45.4±8.2/45.5±8.1 | 32.9 | 0.00 | 0 | UK | DSM-IV | 8 |
| Kuehl(F)(2015) | 28/26 | 41.5±1.8/42.7±2.3 | 25.5 | 0.00 | 50 | Germany | DSM-IV | 8 |
| Kuehl(M)(2015) | 16/15 | 41.3±3.1/38.7±3 | 25.1 | 100.0 | 50 | Germany | DSM-IV | 8 |
| Waloszek(2015) | 25/25 | 16.1±1.4/16.1±1.4 | 22.3 | 24.0 | 0 | USA | DSM-IV | 8 |
| Wingenfeld(2017) | 47/36 | 35.9±11.7/34±12.7 | 22.6 | n/a | 95.8 | Germany | DSM-IV | 7 |
| Roger C. M. Ho(2018) | 61/43 | 37.7±7.6/38.2±9.2 | 23.8 | n/a | 100 | Singapore | DSM-IV | 7 |
| Skibinska(2018) | 30/30 | 38.1±10.2/40.7±11.4 | n/a | n/a | 0 | Poland | DSM-V | 6 |
| Segoviano M.(2018) | 202/206 | 37.3±10/36.8±6.6 | n/a | 16.3 | n/a | Mexico | DSM-V | 7 |
| Eidan,A.J.(2019) | 60/30 | 30.0±13.1/31.1±15.4 | 24.6 | 66.6 | 53.33 | Iraq | DSM-V | 8 |
| Wagner,C.J.(2019) | 130/61 | 46±3.5/42±5.75 | 26.1 | 46.9 | 50.7 | Germany | DSM-IV | 7 |
| Á. Péterfalvi(2019) | 21/20 | 36.1±11.24/35.8±8.53 | 23 | 31.0 | 5 | Hungary | DSM-IV | 8 |
| T. Druzhkova(2019) | 33/43 | 32.9±7.8/30.5±5.5 | 23.7 | n/a | n/a | Russian Federation | ICD-10 | 8 |
| C. Zhang(2020) | 49/50 | 42.3±10.5/42.6±12.3 | 23.3 | n/a | 100 | China | ICD-10 | 7 |
| K. Honkalampi(M) (2021) | 40/112 | 31.3±11.8/49.28±10.27 | 26.3 | 100.0 | 77 | Finland | DSM-IV | 7 |
| K. Honkalampi(F) (2021) | 137/116 | 36.2±10.2/50.2±9.9 | 27.5 | 0 | 77 | Finland | DSM-IV | 8 |
| Y. Liu(2021) | 35/274 | 43.6±15.5/1.6±9.4 | 24.3 | 28.8 | n/a | China | DSM-IV | 8 |
| A. Silić(2022) | 145/148 | 38.6±11.5/39.0±11.0 | 26.2 | 50 | drug-free 3 month | Belgium | ICD-10 | 7 |
| Y. Sánchez-Carro(2022) | 91/80 | 50.6±10.2/49.1±10.2 | 26.3 | 71.4 | 0 | Spain | DSM-IV | 8 |
| R. Yang(2022) | 110/56 | 27.5±8.3/ 29.3± 0.6 | 21.8 | 33 | 0 | China | DSM-IV | 8 |

**Abbreviation:** D/C, depressed and control; DSM, Diagnostic and Statistical Manual of Mental Disorders; CCMD, Chinese Classification of Mental Disorders; ICD-10, international Classification of diseases (10th version); n/a, not available; y, yes; NOS, Newcastle-Ottawa Scale

a. values reflect mean ± SD

b. values determined from three main aspects of selection, comparability and exposure.

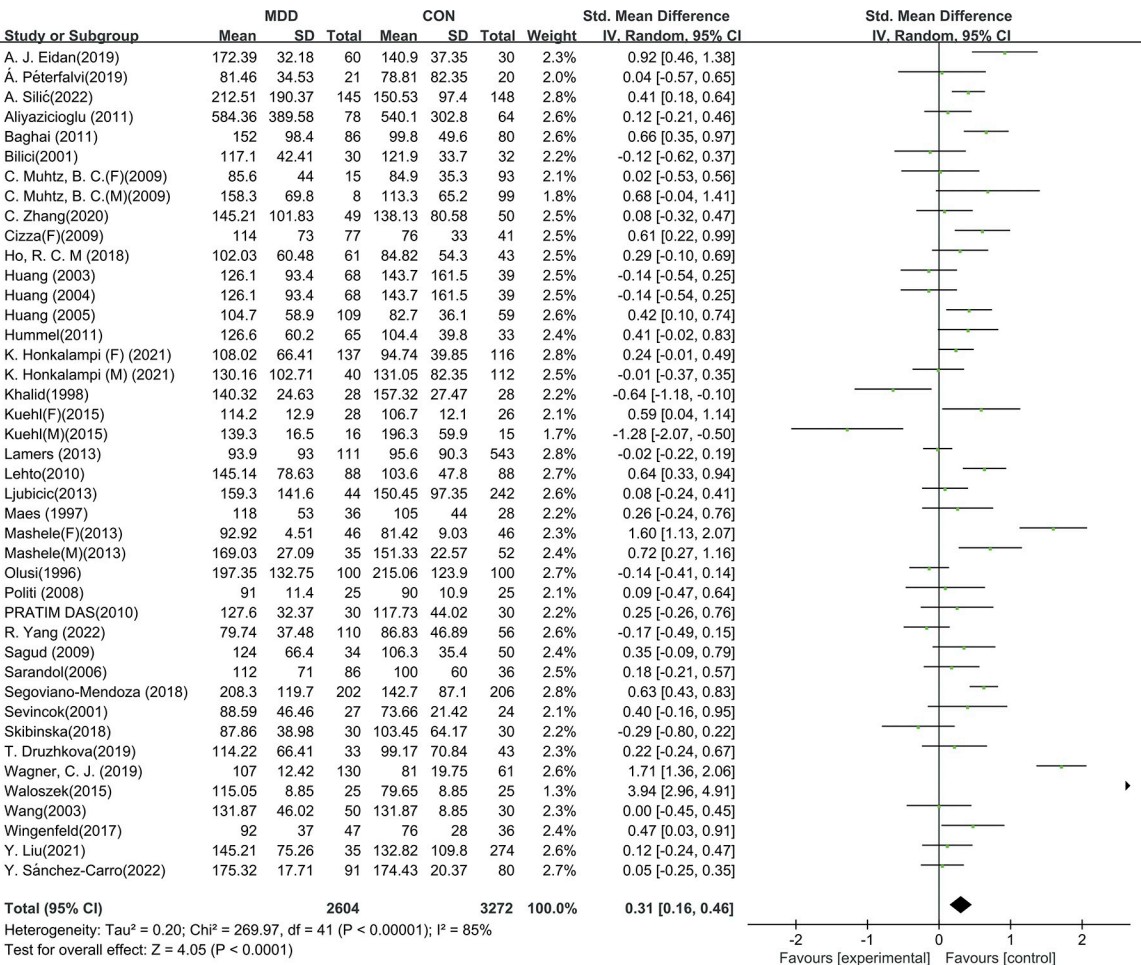

**Fig 2. Comparison of peripheral triglyceride concentration between MDD and HCs.** A total of 2604 MDD patients and of 3272 HCs were applied to meta-analysis. Results demonstrated higher TG concentration in MDD compared to HC through 38 included studies that containing 42 items. Random effects model generated conservative outcomes for the significant heterogeneity. Effect size of each study had relatively equal weight.

(SMD = 0.42, 95% CI: 0.22 to 0.63, $p < 0.01$), and drug-naive (SMD = 0.36, 95% CI: 0.20 to 0.53, $I^2 =$, $p < 0.01$).Differences between the subgroup studied were also conducted, but had no significant difference (age: $\chi^2 = 0.51$, $p = 0.46$;BMI: $\chi^2 = 0.88$, $p = 0.35$; Drug use: $\chi^2 = 0.41$, $p = 0.52$). Subgroup analysis is shown in Table 2 and Figs 4–6.

Meta-regression was used to investigate the effect of sex, age, BMI, diagnosis method, publication date, country and medication on the results. We pooled these variables into meta-regression, calculating with 1000 permutations. The results showed no significant effect (age: $p = 0.47$, diagnosis method: $p = 0.49$, BMI: $p = 0.33$, medication: $p = 0.49$, publication date: $p = 0.41$, country: $p = 0.76$ and sex: $p = 0.50$). Restricted maximum likelihood (REML) estimation of between-study variance demonstrates tau2 = 0.41, and residual variation due to heterogeneity $I^2$ = 78.92% (Table 3). Additionally, all seven variables partly explained heterogeneity in TG levels.

## 4. Discussion

This is the first study to demonstrate high levels of TG in patients with MDD compared to healthy controls, combining meta-regression and subgroup analysis. We included 38 studies

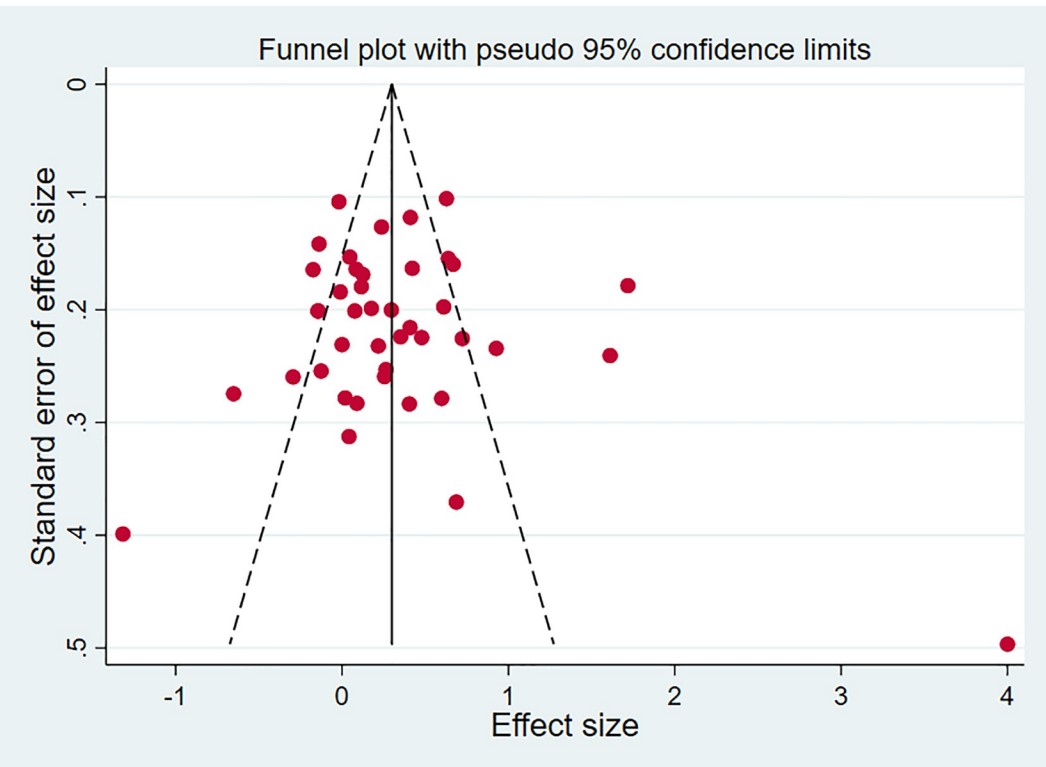

**Fig 3. Funnel plot of publication bias.** The funnel plot was basically symmetrical by visual inspection, following by Egger's test for quantitative analysis and no significant difference found ($p = 0.699$).

with 5876 individuals and demonstrated that TG concentration of patients with MDD was significantly higher than that HCs. Most of the included studies reported that the mean concentrations of TG in patients with MDD and HCs were in the normal range ($< 150$ mg/dl). However, the TG concentration in the MDD group was often near the upper normal limit. Heterogeneity exploration has been applied to age, sex, BMI, antidepressant use publication date, country and diagnosis method through meta-regression analysis, showing no significant results. All seven variables only explained a part of heterogeneity, and there are other potential confounders worth exploring in future studies. Subgroup analysis consistently show similar significant difference in TG levels between patients with MDD and HC. Publication bias is always a concern in meta-analysis. In our study, there was no potential publication bias in all analyses (Egger's test $p > 0.05$).

**Table 2. Subgroup analyses of TG levels in MDD versus HC.**

| Subgroup | | Studies | subjects | SMDs | 95% CI (Conf. Interval) | I² (%) | p value |
|---|---|---|---|---|---|---|---|
| Age | < 45-y | 30 | 4367 | 0.28 | 0.10, 0.47 | 87 | < 0.01 |
| | ≥ 45-y | 12 | 1654 | 0.32 | 0.16, 0.63 | 78 | < 0.01 |
| BMI | < 25 | 17 | 2524 | 0.27 | 0.06, 0.49 | 81 | < 0.01 |
| | ≥ 25 | 19 | 2622 | 0.43 | 0.19, 0.67, | 87 | < 0.01 |
| Medication | using | 18 | 2528 | 0.42 | 0.22, 0.63 | 82 | < 0.01 |
| | naive | 19 | 2559 | 0.36 | 0.20, 0.53 | 86 | < 0.01 |

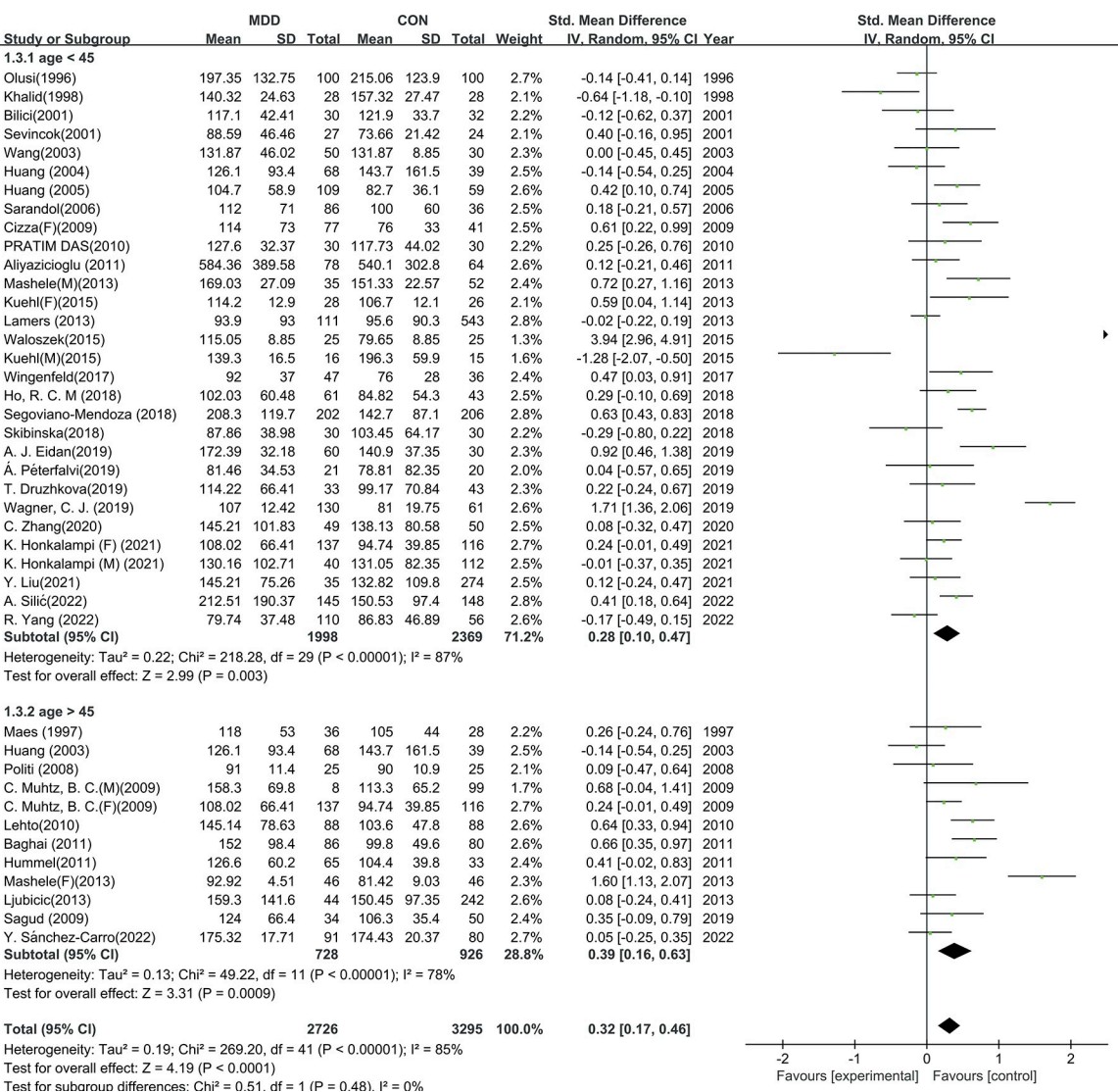

**Fig 4. Subgroup analysis on age.** Forest plots of the TG level between MDD and HC according to age.

Differences in TG levels are associated with fat intake, liver synthesis, and TG catabolism [53]. Although hypertriglyceridemia was more prevalent in patients with MDD than in healthy controls, the mechanisms underpinning the association are poorly understood. The blood TG level reflects the concentration of TG-carrying lipoproteins, including very low-density lipoprotein, VLDL, and chylomicrons. Dietary TG are assembled into chylomicrons in the gut. Their interaction with lipoprotein lipase in the luminal surface of capillary endothelial cells leads to the liberation of free fatty acids, which can pass through cell membranes [54]. Depressed people may have increased appetite due to hyperactivation of putative mesocortico-limbic reward circuitry [55]. Except for four studies, most included studies informed that blood samples were collected after at least 8 hours of fasting, implying that the source of TG was mainly endogenous liver synthesis. TG synthesis is controlled by insulin. In depressed subjects, insulin resistance is a major comorbidity. Insulin resistance leads to unrestrained fat mobilization in adipose tissue, increasing plasma-free fatty acid (FFA) levels. An increased

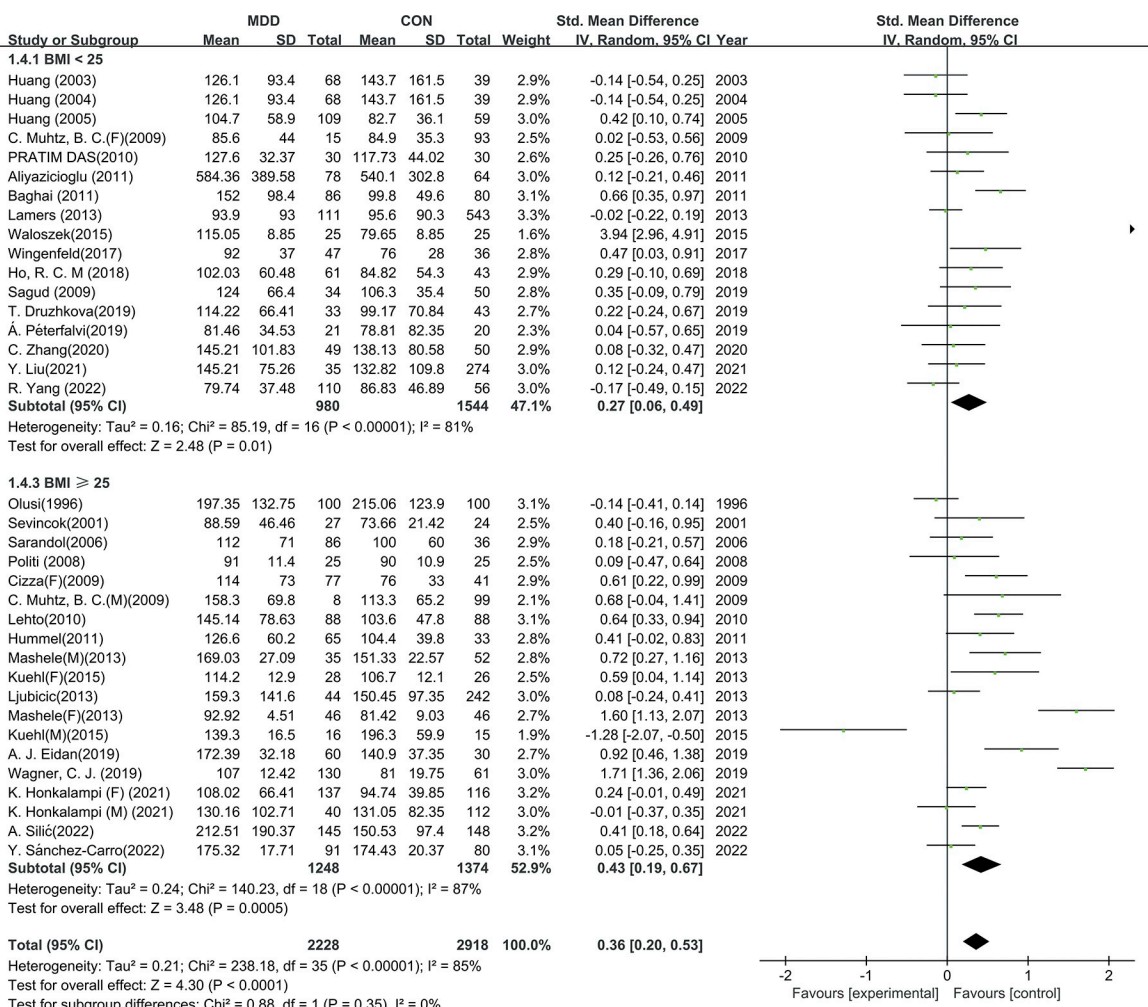

**Fig 5. Subgroup analysis on BMI.** Forest plots of the TG level between MDD and HC according to BMI.

FFA flux into the liver stimulates hepatic lipogenesis and promotes VLDL-TG overproduction [56]. Nevertheless, Wagner *et al.* found that remitted patients with MDD (no depressive episode during the last 12 months) show increased levels of triglyceride [46].

Additionally, the close relationship between depression and cardiovascular disease (CVD) was also clarified in several studies. Robert *et al.* indicated that non-fasting TG is a marker for CVD risk stratification [57], and Van Marwijk *et al.* reported that depression increases the risk of CVD [58]. Besides, other studies revealed that depression significantly increases the risk of stroke, and this increase is probably independent of other risk factors, including hypertension and diabetes [5]. In our study, TG had significantly higher levels in moderate and severe MDD. Although TG itself is not a component of arterial plaque, it is believed that cholesterol within TG-rich particles may contribute to plaque development [59]. Elevated TG level in MDD subjects is probably a risk factor for cardiovascular and cerebrovascular diseases.

This study still has some limitations. Our study contained potential heterogeneity in most analyses, although we adopted a random-effects model, subgroup analysis and meta-regression analyses. The reasons for heterogeneity may be that not all studies reported sufficient demographic data, such as sex, drug use, and BMI, or other factors that could potentially influence

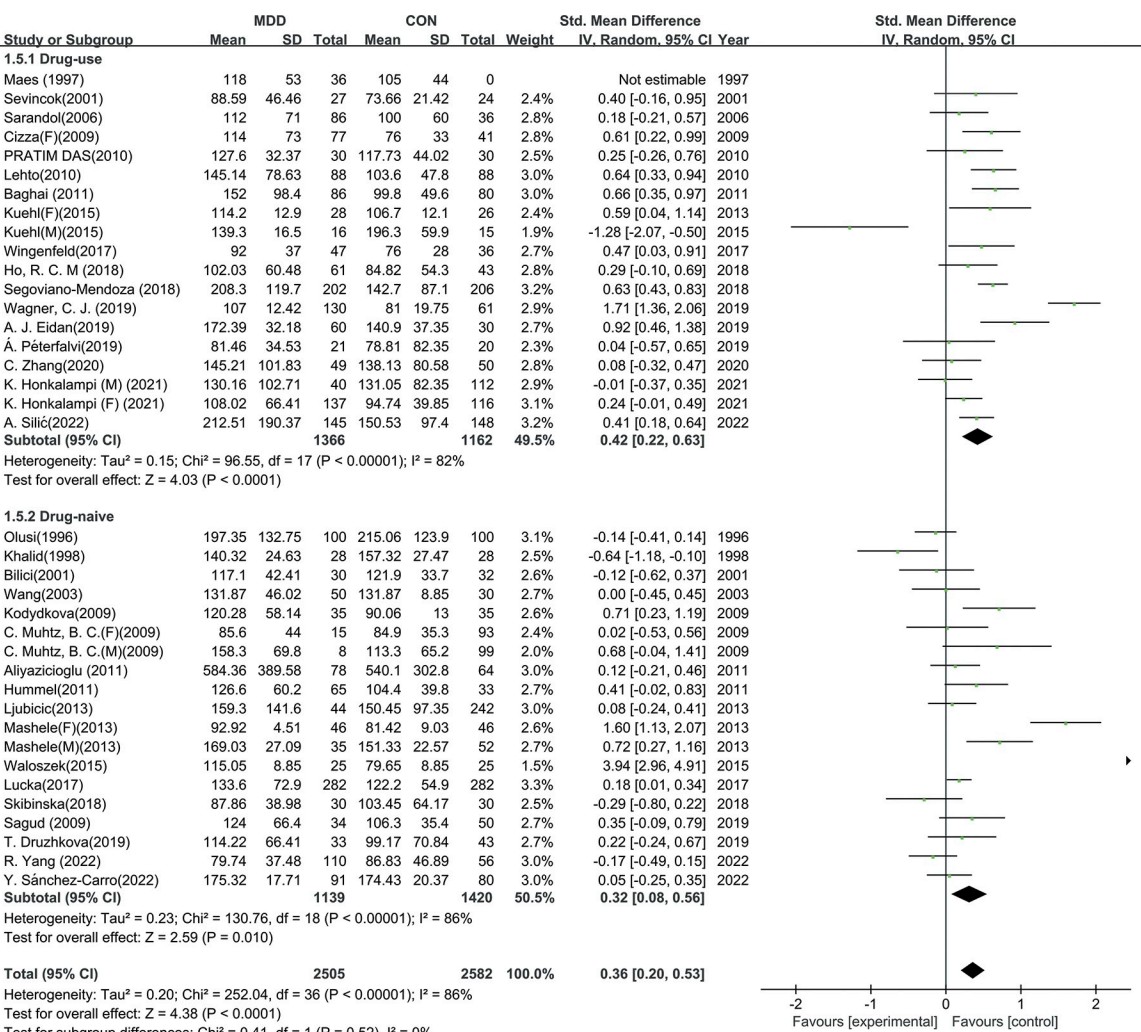

**Fig 6. Subgroup analysis on medication.** Forest plots of the TG level between MDD and HC according to medication.

**Table 3. Meta-regression analysis.**

| Variables | Coef. | Std. Err. | t valuve | 95% CI (Conf. Interval) | p value |
|---|---|---|---|---|---|
| Age | 0.24 | 0.28 | 0.87 | -0.95 to 1.44 | 0.47 |
| Diagnosis | -1.73 | 2.09 | -0.83 | -10.77 to 7.29 | 0.49 |
| BMI | 0.33 | 0.26 | 1.27 | -0.78 to 1.44 | 0.33 |
| Medication | 3.08 | 3.66 | 0.84 | -12.64 to 18.82 | 0.49 |
| Sex | 0.47 | 0.59 | 0.81 | -2.06 to 3.02 | 0.50 |
| Pubdate | -0.21 | 0.20 | -1.02 | -1.08 to 0.66 | 0.41 |
| Country | 0.04 | 0.11 | 0.35 | -0.44 to 0.51 | 0.76 |

**Note:** REML estimate of between-study variance tau2 = 0.41, Residual variation due to heterogeneity $I^2$ = 78.92%. No significant variation for each variable to the SMD of every included studies.

heterogeneity like diet and food intake. Besides, few studies reported the method of assaying TG; thus, TG concentration largely differed among studies. Furthermore, included studies did not adjust the effect of several confounding factors such as genetic factors, alcohol consumption, and cigarette smoking. Additionally, the results of our meta-analysis cannot support the causal effect of depression on TG. A causal association between TG status and depression is biologically plausible. Thus, a risk comparison meta-analysis between depressed and non-depressed subjects is necessary to establish whether higher TG concentrations predict the future development of depression. Finally, our study was not registered in the International Prospective Register of Systematic Reviews (PROSPERO) database, but the protocol was strictly followed the PRISMA guideline items.

## 5. Conclusion

This meta-analytic confirm that depression is associated with elevated concentrations of TG in the peripheral blood. These findings indicate that depression increases the risk of cardiovascular and cerebrovascular diseases. These findings suggest the need to investigate the potential roles of TG in the pathogenesis of depression, identify the potential utility of TG and related lipids biomarkers in monitoring MDD and subsequent stroke or myocardial infarction, and measure potential benefits of low-TG diet in MDD patients.

## Supporting information

**S1 Checklist. PRISMA 2020 checklist.**
(DOCX)

**S1 File. List of the studies that were excluded at the full-text assessment.**
(XLSX)

**S1 Dataset. Data extracted from included studies and used for all analyses.**
(XLSX)

## Acknowledgments

Portion of data in these studies comes from early study "Meta-analysis of peripheral triglyceride of patients with depression", which as our primary studies published in 2014. Besides, the idea of this study inherits from the first author master's thesis named "the evaluation of mental health in bromhidrosis or scars patients".

## Author Contributions

**Data curation:** Di-Ru Xu, Xi Gao.

**Formal analysis:** Ge Tang.

**Methodology:** Di-Ru Xu, Shu-Dong Liu.

**Software:** Ge Tang.

**Supervision:** Yu Chen.

**Validation:** Yu Chen.

**Writing – original draft:** Di-Ru Xu.

**Writing – review & editing:** Li-Bo Zhao, Chan-Juan Zhou, Yu Chen.

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
