## [Decision Letter · Decision Letter 0]

24 Apr 2024

PONE-D-24-10048Association Between Hypertriglyceridemia and Depression : A Systematic Review and Meta-AnalysisPLOS ONE

Dear Dr. yu,

Thank you for submitting your manuscript to PLOS ONE. After careful consideration, we feel that it has merit but does not fully meet PLOS ONE’s publication criteria as it currently stands. Therefore, we invite you to submit a revised version of the manuscript that addresses the points raised during the review process.

We look forward to receiving your revised manuscript.

Kind regards,

Michele Fornaro

Academic Editor

PLOS ONE

Journal Requirements:

"National Natural Science Foundation of China (Grant No. 81601207)"

"..This work was supported by the National Natural Science Foundation of China (Grant No. 81601207).

The authors would like to express their gratitude to EditSprings (https://www.editsprings.cn ) for the expert linguistic services provided."

Please note that funding information should not appear in the Acknowledgments section or other areas of your manuscript. We will only publish funding information present in the Funding Statement section of the online submission form. Please remove any funding-related text from the manuscript. 

4. Please upload a copy of Figures 4 to 7, to which you refer in your text on page 11. If the figure is no longer to be included as part of the submission please remove all reference to it within the text.

Reviewers' comments:

Reviewer's Responses to Questions

**Comments to the Author**

1. Does the manuscript provide a valid rationale for the proposed study, with clearly identified and justified research questions?

Reviewer #1: Yes

Reviewer #2: Yes

2. Is the protocol technically sound and planned in a manner that will lead to a meaningful outcome and allow testing the stated hypotheses?

Reviewer #1: No

Reviewer #2: No

3. Is the methodology feasible and described in sufficient detail to allow the work to be replicable?

Reviewer #1: Yes

Reviewer #2: Yes

4. Have the authors described where all data underlying the findings will be made available when the study is complete?

Reviewer #1: No

Reviewer #2: Yes

5. Is the manuscript presented in an intelligible fashion and written in standard English?

Reviewer #1: Yes

Reviewer #2: Yes

6. Review Comments to the Author

You may also provide optional suggestions and comments to authors that they might find helpful in planning their study.

Reviewer #1: This manuscript by Di-Ru Xu and colleagues aims to synthesise current evidence on the differences in triglyceride (TG) concentrations between patients with major depressive disorder (MDD) and healthy controls (HCs), by conducting a systematic review and meta-analysis (MA).

This is a very relevant topic for clinical psychiatry given the close relationship between mental disorders, metabolic alterations, cardiovascular risk, and mortality.

Nevertheless, the MA presents some methodological issues that need to be taken into consideration before publication.

MAYOR COMMENT:

The inclusion criteria are not clear: On page 4, Section 2.2 Study selection, you define "3) inclusion of a depressed group diagnosed by DSM (III/IV/V), CCMD, or ICD-10," however, it is not clear whether only patients with MDD are included or also other diagnoses. In Section 2.1 Search strategy, you include terms such as "depressive symptom" and "dysthymia" in your search string. Please clarify this point. If diagnoses other than MDD are included, it is necessary to modify the entire manuscript to refer to "depressive disorders" rather than MDD. Additionally, in Table 1, the diagnoses made in each study should be reported (this is probably another source of heterogeneity?).

The comparator is not clear either. On page 4, Section 2.2 Study selection, you define "3) inclusion of a psychiatrically healthy group." Apart from the fact that this should be criterion number 4, there are some issues with this definition: 1) "Psychiatrically" implies that patients may have medical disorders? This could be another source of heterogeneity. Please clarify this point, and if patients have medical disorders, report them in Table 1. In any case, avoid using the term "psychiatrically." How were healthy controls defined (DSM/ICD?, none?)?

It's not clear how heterogeneity exploration was handled.

The authors report having conducted both meta-regression and subgroup analysis for age, gender, and BMI. While it may make sense to perform both subgroup analysis and meta-regression for the same variables in a meta-analysis, it's important to consider a few issues:

1) Analysis objective: Subgroup analysis and meta-regression serve different purposes. Particularly for the age variable, I don't understand the rationale behind conducting subgroup analysis by dividing into age < or > 45 years. Are there any specific evidences suggesting triglyceride levels change specifically above 45 years? Also, how were these 45 years considered? Based on the mean age of the studies or inclusion criteria of the primary studies? The analysis should be conducted based on the latter. Please reconsider conducting subgroup analysis, favoring meta-regression. For BMI, I think it may make sense to conduct both meta-regression and subgroup analysis because studying the difference between normal-weight and overweight patients makes sense (unlike age 45), but the same considerations regarding mean of the studies or inclusion criteria apply. Specify it.

2) Data availability regarding sex: If you have access to the number of males and females, and thus the percentage of males and females, it makes much more sense to conduct only a meta-regression that includes many studies, rather than a subgroup with few studies and therefore less reliable results. Please consider conducting only meta-regression and not subgroup analysis (only 7 studies out of 46). In general, it is always better to conduct further analyses when you have at least 10 studies.

Furthermore, the purpose of a subgroup analysis is also to test the difference between the identified groups (Test for subgroup differences). The authors only report the effect sizes for the two subgroups without stating whether these differences are significant based on the test. Therefore, it cannot be concluded that the variable (age, BMI, medication use, sex) significantly moderates the effect. Please report the Test for subgroup differences. Indeed, the directions of the effect do not seem to be very different, and there may be no differences between the groups studied. This would be compatible and consistent with the results of the meta-regressions.

3) In addition to these analyses, other variables should be taken into consideration. The effect of medication use has only been considered based on whether patients are taking medication or not. Another important factor is which medications they are taking. It's not uncommon for major depressive disorder patients to be prescribed medications such as second-generation antipsychotics (olanzapine, quetiapine) that are associated with metabolic alterations. Was it possible to consider this variable? Other important variables would include the study date and the country of the study. There could be year and regional differences to take into account.

Plase, conduct these analyses.

Study degign: Why did you include only observational studies and not RCTs?

Please provide the protocol registration number.

MINOR MINOR COMMENTS:

The word 'hypertriglyceridemia' appears in the title. I don't think it's correct. The meta-analysis compares triglyceride levels between MDD and HCs. As the authors themselves emphasize in the discussion, the values often fall within normal ranges in both groups.

In the abstract, it is redundantly repeated that meta-regressions and subgroup analyses were conducted to investigate heterogeneity.The use of the word "covariate" in reference to meta-regressions is not correct.

In both the abstract and results sections, why use the word "approximately"? It's not approximate; it's your result. Just report the effect size and the 95% CI.

Throughout the manuscript, please use the word "sex" (biologically determined) instead of "gender" (socially and culturally determined).

Among the inclusion criteria, you only mention ICD-10 for diagnosis and not other versions. Why? Did you find only this, or was a diagnosis with ICD-9 not considered? I believe it's the former, but the criteria should be a priori.

You mentioned that data were not available for 33 studies. In these cases, did you reach out to the authors?

Reviewer #2: Dear all,

Thank you for sending the paper titled "Association Between Hypertriglyceridemia and Depression: A Systematic Review and Meta-Analysis" and for allowing me to review it. The authors conducted a meta-analysis to evaluate the disparity in blood triglyceride concentrations between individuals with depression and healthy controls. They concluded that triglyceride levels were elevated in those with depression compared to the control group.

Below I will detail my concerns and/or suggestions to improve this paper:

1. Could you please provide a clearer definition of your study population (cases)? It appears that you are interested in examining individuals with major depressive disorder (as defined by DSM, ICD, or CCD criteria). However, it's not explicitly stated whether your inclusion criteria encompassed only those with a confirmed diagnosis of major depressive disorder, or if you also intended to include individuals with depressive symptoms or dysthymia (as indicated in the search string).

2. I don’t understand your exclusion criteria depressive symptoms in the context of medical illnesses (e.g., hyperlipemia, dyslipidemia). Are you interested in observing the differences in tryglicerides between people without a diagnosis of hypertriglyceridemia? Please, specify.

3. The authors used the mean difference as the effect size, which offers clinical insight into the disparity in triglyceride levels between the two populations. However, I believe it would be useful to also include a standardized mean difference, which could facilitate comparison of your findings with those of other studies.

4. Please, provide in the supplemental material a list of the studies that were excluded at the full-text assessment, with reasons, as required by PRISMA guidelines.

5. Did you have a prepublished protocol? If not, this should be considered as an important limitation.

6. The cut-offs chosen for subgroup analyses seem to be quite arbitrary (especially without a protocol). Please, provide the reasons behind choosing these specific cut-offs.

7. The authors present both subgroup analyses and meta-regressions. However, both subgroup analyses and meta-regressions follow the same underlying logic. Furthermore, in their reporting of the subgroup analysis regarding age (comparing individuals aged more and less than 45 years), they omit the comparison statistic between the two subgroups, which, according to the meta-regression, should not be significant. I suggest to revise this aspect and consistently include a statistic comparing the two subgroups, or alternatively, solely report the meta-regressions.

I hope that these suggestions may help improving the manuscript.

Best wishes,

Reviewer

7. PLOS authors have the option to publish the peer review history of their article (what does this mean?). If published, this will include your full peer review and any attached files.

Reviewer #1: No

Reviewer #2: No

---

## [Author Response · Author response to Decision Letter 0]

28 May 2024

Dear Editor and Reviewers:

We would like to express our sincere appreciation to you and the reviewers for their valuable comments and suggestions on our manuscript titled " Association between triglyceride and depression: A systematic review and meta-analysis”. We are grateful for the time and effort invested in reviewing our work. We have carefully considered all the comments and have made revisions accordingly. in response to the reviewer's comments, we would like to provide a point-by-point response:

Editor:

Comment 1: “Please ensure that your manuscript meets PLOS ONE's style requirements, including those for file naming”

Response 1: We had revised the format of manuscript as the requirements of PLOS ONE's, that performing through the style templates provided in https://journals.plos.org/plosone/s/file?id=wjVg/PLOSOne_formatting_sample_main_body.pdf and https://journals.plos.org/plosone/s/file?id=ba62/PLOSOne_formatting_sample_title_authors_affiliations.pdf

Comment 2: “Please state what role the funders took in the study” and “Please include this amended role of funder statement in your cover letter”

Response 2: We revised acknowledgements of “The funder of Professor Li Bo Zhao had proofread the manuscript” in cover letter already.

Comment 3: “Please note that funding information should not appear in the Acknowledgments section or other areas of your manuscript. We will only publish funding information present in the Funding Statement section of the online submission form. Please remove any funding-related text from the manuscript.”

Response 3: We had deleted “This work was supported by the National Natural Science Foundation of China (Grant No. 81601207)” and “The authors would like to express their gratitude to EditSprings (https://www.editsprings.cn ) for the expert linguistic services provided” in manuscript. This statement will present in the Funding Statement section of the online submission form.

Comment 4: “Please upload a copy of Figures 4 to 7, to which you refer in your text on page 11. If the figure is no longer to be included as part of the submission, please remove all reference to it within the text”

Response 4: We uploaded figure 4 to 6. As requirement of reviewers, subgroup analysis in sex was deleted, so only three subgroup analysis were conducted (results demonstrated in Fig 4 to 6).

Comment 5: “Please include captions for your Supporting Information files at the end of your manuscript, and update any in-text citations to match accordingly”

Response 5: We followed the Supporting Information guidelines on http://journals.plos.org/plosone/s/supporting-information, including captions at the end of our manuscript (named as S1 Checklist, S2 File and S3 Dataset).

Reviewer #1

MAYOR COMMENT:

Comment 1: “The inclusion criteria are not clear: On page 4, Section 2.2 Study selection, you define "3) inclusion of a depressed group diagnosed by DSM (III/IV/V), CCMD, or ICD-10," however, it is not clear whether only patients with MDD are included or also other diagnoses.”

Response 1: We revised the description of inclusion criteria as “Studies that patients only diagnosed MDD”, that we actually performed in study selection.

Comment 2: “In Section 2.1 Search strategy, you include terms such as "depressive symptom" and "dysthymia" in your search string. Please clarify this point. If diagnoses other than MDD are included, it is necessary to modify the entire manuscript to refer to "depressive disorders" rather than MDD. Additionally, in Table 1, the diagnoses made in each study should be reported (this is probably another source of heterogeneity?)”

Response 2: When we draw up to the protocol of search strategy previously, in order to get more outcomes, so "depressive symptom" and "dysthymia" were included. But all included studies were strictly diagnosed as MDD through applicable standards, such as DSMs, ICDs and CCMDs. The diagnose of each study was reported in table 1. Different diagnose method was performed in searching source of heterogeneity using meta-regression subsequently.

Comment 3: “The comparator is not clear either. On page 4, Section 2.2 Study selection, you define "3) inclusion of a psychiatrically healthy group." Apart from the fact that this should be criterion number 4, there are some issues with this definition: 1) "Psychiatrically" implies that patients may have medical disorders? This could be another source of heterogeneity. Please clarify this point, and if patients have medical disorders, report them in Table 1. In any case, avoid using the term "psychiatrically." How were healthy controls defined (DSM/ICD? none?)?”

Response 3: First, we revised “criterion number 3” to “criterion number 4”. Then, the healthy groups in these included studies were defined as diagnose without any disease through applicable standards screening and routine medical examination when we browsing literature content, so we revised description of criterion number 4 to “healthy controls defined as those who are not diagnosed with any disease”. We also examined previous included studies, found 8 studies reported combining medical illness, such as diabetes et al., Meanwhile these studies had deleted and statistic outcomes were following changed.

Comment 4: “It's not clear how heterogeneity exploration was handled. 1) Analysis objective: Subgroup analysis and meta-regression serve different purposes. Particularly for the age variable, I don't understand the rationale behind conducting subgroup analysis by dividing into age < or > 45 years. Are there any specific evidences suggesting triglyceride levels change specifically above 45 years? Also, how were these 45 years considered? Based on the mean age of the studies or inclusion criteria of the primary studies? The analysis should be conducted based on the latter. Please reconsider conducting subgroup analysis, favoring meta-regression. For BMI, I think it may make sense to conduct both meta-regression and subgroup analysis because studying the difference between normal-weight and overweight patients makes sense (unlike age 45), but the same considerations regarding mean of the studies or inclusion criteria apply. Specify it.”

Response 4: We stratiﬁed age < and > 45 years based on primary study, which reported young patient (< 45 years) with ST-elevation myocardial infarction (STEMI) has significant higher triglyceride and Lipoprotein(a) level compared to middle-old age patients (> 45). Additionally, some other studies researching schizophrenia also stratiﬁed age by 45 years[1]. This stratificational evidence had supplied in “2.4 Statistical analysis” section of our manuscript. For BMI, stratificational criteria was supplied as well by normal-weight and overweight (> 25 or < 25).

Comment 5: “2) Data availability regarding sex: If you have access to the number of males and females, and thus the percentage of males and females, it makes much more sense to conduct only a meta-regression that includes many studies, rather than a subgroup with few studies and therefore less reliable results. Please consider conducting only meta-regression and not subgroup analysis (only 7 studies out of 46). In general, it is always better to conduct further analyses when you have at least 10 studies.”

Response 5: After considering this issue carefully, the outcome of subgroup analysis for sex was not reliable enough. So we canceled the subgroup analysis for sex, only conducting meta-regression.

Comment 6: “Furthermore, the purpose of a subgroup analysis is also to test the difference between the identified groups (Test for subgroup differences). The authors only report the effect sizes for the two subgroups without stating whether these differences are significant based on the test. Therefore, it cannot be concluded that the variable (age, BMI, medication use, sex) significantly moderates the effect. Please report the Test for subgroup differences. Indeed, the directions of the effect do not seem to be very different, and there may be no differences between the groups studied. This would be compatible and consistent with the results of the meta-regressions.”

Response 6: We had supplied the testing outcome for subgroup differences in line 113 to 115. There was no differences between the groups studied that being compatible and consistent with the results of the meta-regressions (p ＞ 0.05).

Comment 7: “3) In addition to these analyses, other variables should be taken into consideration. The effect of medication use has only been considered based on whether patients are taking medication or not. Another important factor is which medications they are taking. It's not uncommon for major depressive disorder patients to be prescribed medications such as second-generation antipsychotics (olanzapine, quetiapine) that are associated with metabolic alterations. Was it possible to consider this variable? Other important variables would include the study date and the country of the study. There could be year and regional differences to take into account. Please, conduct these analyses.”

Response 7: We also tried to clarify which antidepressants patients are taking, even whether second-generation antipsychotics had applied. But most studies had not reported this detail, leading our analysis can not conduct for this variable. In meta-regression, we conducted publication year and country as variables, outcomes reporting in Table 3. Because fasting procession was performed in the including criteria, so the variable of fast had been deleted in meta-regression analysis.

Comment 8: “Study design: Why did you include only observational studies and not RCTs?”

Response 8: Because most RCT studies were designed for intervention study, such as clinical drug research. This point was not fit the purpose of our study, so RCTs were not considered.

Comment 9: “Please provide the protocol registration number.”

Response 9: Unfortunately, our study was not registered in the International Prospective Register of Systematic Reviews (PROSPERO) database, but the protocol was strictly followed the PRISMA guideline items. This limitation had descripted in line 161 to 162.

MINOR MINOR COMMENTS:

Comment 10: “The word 'hypertriglyceridemia' appears in the title. I don't think it's correct. The meta-analysis compares triglyceride levels between MDD and HCs. As the authors themselves emphasize in the discussion, the values often fall within normal ranges in both groups.”

Response 10: We revised title from “hypertriglyceridemia” to “triglyceride”.

Comment 11: “In the abstract, it is redundantly repeated that meta-regressions and subgroup analyses were conducted to investigate heterogeneity. The use of the word "covariate" in reference to meta-regressions is not correct.”

Response 11: The redundant portion had been deleted. And "covariate" in reference was changed to “variables”.

Comment 12: “In both the abstract and results sections, why use the word "approximately"? It's not approximate; it's your result. Just report the effect size and the 95% CI.”

Response 12: We revised reporting the specific value of effect size and 95% CI in the abstract and results sections.

Comment 13: “Throughout the manuscript, please use the word "sex" (biologically determined) instead of "gender" (socially and culturally determined).”

Response 13: We revised all description of gender into sex throughout the manuscript.

Comment 14: “Among the inclusion criteria, you only mention ICD-10 for diagnosis and not other versions. Why? Did you find only this, or was a diagnosis with ICD-9 not considered? I believe it's the former, but the criteria should be a priori.”

Response 14: In our searching procession, any edition of ICDs, even including DMSs and CCMDs were considered, but none of other edition ICDs except ICD-10 study had been found. So we revised the description to “Studies that patients only diagnosed MDD on any edition of ……” in the section of 2.2 Study selection, line 48.

Comment 15: “You mentioned that data were not available for 33 studies. In these cases, did you reach out to the authors?”

Response 15: Sorry, we had not reach out to contact the authors of these studies.

Reviewer #2:

Comment 1: “Could you please provide a clearer definition of your study population (cases)? It appears that you are interested in examining individuals with major depressive disorder (as defined by DSM, ICD, or CCD criteria). However, it's not explicitly stated whether your inclusion criteria encompassed only those with a confirmed diagnosis of major depressive disorder, or if you also intended to include individuals with depressive symptoms or dysthymia (as indicated in the search string)”

Response 1: All included studies were strictly diagnosed only as MDD through applicable standards, such as DSMs, ICDs and CCMDs. The diagnose of each study was reported in table 1. So, we revised the description of inclusion criteria as “Studies that patients only diagnosed MDD”. “Depressive symptoms or dysthymia” in our search strategy just to increase the number of potential target literature, to witch avoid missing references that could be included.

Comment 2: “I don’t understand your exclusion criteria depressive symptoms in the context of medical illnesses (e.g., hyperlipemia, dyslipidemia). Are you interested in observing the differences in triglycerides between people without a diagnosis of hypertriglyceridemia? Please, specify.”

Response 2: In our excluding criteria, MDD patients having medical illnesses such as hyperlipemia or dyslipidemia et al before first onset of depression were excluded. This strategy was conducted to avoid the outcome of triglyceride levels disturbed by other medical illnesses. Thus, we had revised the description of inclusion criteria as “2) medical illnesses before onset of MDD (e.g., hyperlipemia, dyslipidemia, diabetes mellitus et al” in line 51.

Comment 3: “The authors used the mean difference as the effect size, which offers clinical insight into the disparity in triglyceride levels between the two populations. However, I believe it would be useful to also include a standardized mean difference, which could facilitate comparison of your findings with those of other studies.”

Response 3: We carefully consider the analysis of effect size. The different measurement for triglyceride, even different table of weight and measures, to avoid the outcome affected by baseline risk and has better consistency, we conducted calculation as SMD (standardized mean difference) instead of WMD (weighted mean difference).

Comment 4: “Please, provide in the supplemental material a list of the studies that were excluded at the full-text assessment, with reasons, as required by PRISMA guidelines”

Response 4: At last, 108 studies were excluded for different reasons, such as data not available, no control groups, combined other disease and et al. We had produced a excel form, as a supplemental material named “S2 file”, for demonstrating the reasons of excluding full-text assessment. 

Comment 5: “Did you have a republished protocol? If not, this should be considered as an important limitation”

Response 5: Unfortunately, our study was not registered in the International Prospective Register of Systematic Reviews (PROSPERO) database, but the protocol was strictly followed the PRISMA guideline items. This limitation had descripted in line 161 to 162.

Comment 6: “The cut-offs chosen for subgroup analyses seem to be quite arbitrary (especially without a protocol). Please, provide the reasons behind choosing these specific cut-offs.”

Response 6: We stratiﬁed age < and > 45 years based on primary study, which reported young patient (< 45 years) with ST-elevation myocardial infarction (STEMI) has significant higher triglyceride and Lipoprotein(a) level compared to middle-old age patients (> 45). Additionally, some other studies researching schizophrenia also stratiﬁed age by 45 years[1]. This stratificational evidence had supplied in “2.4 Statistical analysis” section of our manuscript. For BMI, stratificational criteria was supplied as well by normal-weight and overweight (> 25 or < 25). For drug use, drug-naive patients and antidepressant used patients had divided into subgroup for comparative analysis.

Comment 1: “The 

---

## [Decision Letter · Decision Letter 1]

23 Sep 2024

Association Between Triglyceride and Depression: A systematic Review and Meta-analysis

PONE-D-24-10048R1

Dear Dr. yu,

We’re pleased to inform you that your manuscript has been judged scientifically suitable for publication and will be formally accepted for publication once it meets all outstanding technical requirements.

Kind regards,

Md. Rabiul Islam, PhD

Academic Editor

PLOS ONE

Additional Editor Comments (optional):

I've reviewed your revised manuscript and the comments from the reviewers, and I think all of the concerns raised have been satisfactorily addressed.

Reviewers' comments:

Reviewer's Responses to Questions

**Comments to the Author**

1. Does the manuscript provide a valid rationale for the proposed study, with clearly identified and justified research questions?

Reviewer #1: Yes

Reviewer #3: Yes

2. Is the protocol technically sound and planned in a manner that will lead to a meaningful outcome and allow testing the stated hypotheses?

Reviewer #1: Yes

Reviewer #3: Yes

3. Is the methodology feasible and described in sufficient detail to allow the work to be replicable?

Reviewer #1: Yes

Reviewer #3: Yes

4. Have the authors described where all data underlying the findings will be made available when the study is complete?

Reviewer #1: Yes

Reviewer #3: Yes

5. Is the manuscript presented in an intelligible fashion and written in standard English?

Reviewer #1: Yes

Reviewer #3: Yes

6. Review Comments to the Author

You may also provide optional suggestions and comments to authors that they might find helpful in planning their study.

Reviewer #1: I thank the authors for adequately responding to my review, I have no more comments and I reccomend the manoscript for publication

Reviewer #3: The author answered all the reviewer comments carefully. They corrected their manuscript accordingly. I think we can accept this paper for publication.

7. PLOS authors have the option to publish the peer review history of their article (what does this mean?). If published, this will include your full peer review and any attached files.

Reviewer #1: No

Reviewer #3: No

---

## [Editor Report · Acceptance letter]

25 Sep 2024

PONE-D-24-10048R1 

PLOS ONE

Dear Dr. Chen, 

I'm pleased to inform you that your manuscript has been deemed suitable for publication in PLOS ONE. Congratulations! Your manuscript is now being handed over to our production team.

Kind regards, 

on behalf of

Dr. Md. Rabiul Islam 

Academic Editor

PLOS ONE